

# Estimate of lysine nutritional requirements for Japanese quail breeders

Lizia Cordeiro de Carvalho[1], Manoela Garcia Borgi Lino de Sousa[1], Jaqueline Aparecida Pavanini[1], Tadia Emanuele Stivanin[2], Nelson José Peruzzi[1], Alan Rodrigo Panosso[1], Michele Bernardino de Lima[1] and Edney Pereira da Silva[1]

[1] Universidade Estadual Paulista, Department of Animal Science, College of Agriculture and Veterinary Sciences, Jaboticabal, São Paulo, Brasil
[2] Genetic improvement of Japanese Quails, Vicami Codornas, Assis, São Paulo, Brasil

## ABSTRACT

**Background**. Japanese quail breeders are the basis for genetic improvement and multiplication for commercial layers, however, there have been no known studies on the optimal lysine level for these birds. Thus, study the egg output response to the lysine (Lys) supply using different e-functions and evaluate the that best fit, have allowed the partition the lysine requirements for maintenance, both weight and egg output maximum.

**Methods**. The objectives of this study were to identify the responses to various Lys levels, identify the functions related to these responses and determine the ideal Lys intake amount for Japanese quail breeders. A completely randomized design of seven treatments with seven replicated was used. Treatments consisted of diet supplementation by Lys in concentrations of 16.8, 11.8, 8.4, 6.7, 5.0, 3.4, and 1.7 g/kg. Six exponential models were adjusted.

**Results**. The level of Lys was found to affect bird responses ($P < 0.001$). The birds responded to the levels provided, allowing for the creation of a lysine response curve. A monomolecular function with four parameters was balanced against the statistics of adjustment and selection of models. It was possible to estimate the level of lysine required for maintenance as $133 \pm 2$ mg/kg BW$^{0.67}$, and based an average of 41% efficiency, 22 mg Lys produced 1 g of egg output (EO). The daily intake calculated by the monomolecular factorial model was 284 mg Lys for a bird with 0.170 kg body weight and production of 10 g EO/day. The four-parameter monomolecular function proposed in this study is adequate for interpreting the animal response and calculating lysine intake for breeders.

## INTRODUCTION

Japanese quail breeders is the basis for genetic improvement and multiplication for commercial layers. Through the selection process, in each generation, genetically superior animals are used to form the breeding stock. Approximately 4,444 breeder birds are required to produce one million Japanese quails, and in Brazil, more than 111,000 were necessary to yield the 25 million currently housed (*Silva et al., 2020*). The breeders used in

Corresponding author
Edney Pereira da Silva,
edney.silva@unesp.br

this study were selected for egg production and belong to the male line of laying quails. There have been no known studies on the optimal amino acid level for Japanese quails breeders. Lysine (Lys) is the reference amino acid for establishing the ideal relationship, and it is the second limiting amino acid in maize and soybean diets of birds. Lys acts on protein and lipid metabolism, and in reproduction, subdoses are related to atrophy of reproductive organs and the liver (*Ruan et al., 2019*; *Tian et al., 2019*).

Among the main methods used to establish amino acid intake are dose response and factorial analysis. The factorial method is a reasonable option to establish amino acid intake by using variables such as body weight (BW) and egg output (EO) (*Silva et al., 2019*). This method is based on using the linear relationships between amino acid intake and these variables (mg per kg BW and mg per g EO) to partition the BW maintenance requirement and EO production (*Sakomura et al., 2015*; *Reis et al., 2018*). A reduction in feed conversion (*Basaglia et al., 2005*) is possible because the input values for BW and EO correspond to the average population potential (*Hauschild, Pomar & Lovatto, 2010*).

Although feed conversion reduction in the egg production industry can be useful, the effectiveness of this reduction is based on the average production potential of the batch (*Basaglia et al., 2005*; *Silva et al., 2015a*), and thus individuals above the average population potential would inevitably receive a subdose (*Hauschild, Pomar & Lovatto, 2010*; *da Silva et al., 2019*; *Silva et al., 2015a*). This characteristic of the factorial method may be a limitation for nutritionists in genetic improvement programs and multiplication systems, in terms of how individuals performing above the population average should be adequately identified and nurtured. Therefore, since linear relationships are limited because they are infinite in all directions, empirical constraints should be used to obtain more precise estimates that represent a closer approximation to the actual condition (*Silva et al., 2015a*; *da Silva et al., 2019*).

Nonlinear factorial models are alternatives for breeders (*Kebreab et al., 2008*; *Ekmay et al., 2014*) especially for birds that prioritize reproduction by mobilizing body reserves to maintain egg laying (*Lima et al., 2018*; *Lima et al., 2020*). Exponential functions allow for the consideration of maintenance and production partitioning (*Samadi Liebert, 2008*; *Dorigam et al., 2017*) while catering to the most productive animals of the population, since curvilinear adjustment can change the response rate ($\alpha/\beta$ x) with the approximation of the maximum genetic potential (*Fuller & Garthwaite, 1993*; *da Silva et al., 2019*). Factorial models based on e-functions are available that have parameters with biological significance that can be improved, such as the requirement to maintain unity on the axis of the ordinate (*Samadi Liebert, 2008*; *Dorigam et al., 2014*; *Dorigam et al., 2017*) when the ideal would be on the axis of the abscissa (*Kebreab et al., 2008*; *Sarcinelli et al., 2020*; *Silva et al., 2020*), thereby avoiding confusion between the minimal response on the ordinate axis (*Lima et al., 2013*; *Dorigam et al., 2017*) and the requirement of nutrient maintenance on the abscissa axis. Therefore, this study aimed to (1) study the EO response to the Lys supply using different e-functions, (2) evaluate the e-functions that best fit the EO responses, (3) partition the Lys requirements for BW maintenance and EO production, and (4) the Lys intake level that maximizes EO.

## MATERIALS & METHODS

The study was conducted in the Laboratory of Poultry Sciences of the Departament of Zootecnia da Faculdade de Ciências Agrárias e Veterinárias of the Universidade Estadual Paulista, Campus of Jaboticabal, São Paulo, Brazil. The procedures used in this study were approved by the Committee on Animal Use Ethics, under protocol 012203/17.

### Birds, housing, and experimental design

Forty-nine VICAMI® Japanese quails breeders were used at 14 weeks of age, when they are at their peak performance. The experiment was conducted in a temperature-controlled climate chamber containing galvanized wire cages measuring 0.26 m × 0.37 m × 0.36 m, with channel feeders and nipple drinkers. The temperature during the experimental period was maintained at 24 °C, with a 16:8 h (L:D) photoperiod. Water was provided ad libitum. A completely randomized design was used, with seven treatments and seven repetitions. Each experimental unit consisted of one bird per cage. All cages were identified with different colored labels, according to the treatments. The treatments consisted of seven levels of Lys in the diet as follows: D7–16.8 g/kg; D6–11.8 g/kg; D5–8.4 g/kg; D4–6.7 g/kg; D3–5.0 g/kg; D2–3.4 g/kg, and D1–1.7 g/kg. After the trial the animals remained in the university's herd for egg production.

### Experimental treatments and diets

The level of Lys in the dietary protein profile and experimental diets were formulated as described by *Fisher & Morris (1970)*. A formulation with a high crude protein content (HPD) and a relative deficiency in Lys compared to the other amino acids, and a second formulation that was free of protein and amino acids (NFD) were prepared (Table 1). The nutritional levels of the essential amino acids in the HPD were based on the recommendations described previously by *Rostagno et al. (2011)* (Table 2). The Lys level was established by multiplying the recommended amount by 1.5, and that of the other amino acids by 2.0 to maintain a minimum Lys deficiency of 50% compared to the other amino acids. For energy and other nutrients (vitamins and minerals), the minimum recommendations were followed by *Rostagno et al. (2011)*. NFD was formulated to provide energy and the other nutrients with no amino acids. The intermediate experimental levels of Lys were obtained by diluting the HPD with NFD in the following proportions (HPD:NFD): 100:0; 70:30; 50.1:49.9; 40:60; 30:70; 20.1:79.9; and 10:90; thus obtaining Lys concentrations of 16.8, 11.8, 8.4, 6.7, 5.0, 3.4, and 1.7 g/kg respectively.

### Measurements and variables analysed

The experiment occurred for 22 days, with the first 7 days of adaptation. The feed supply was according to the body weight (measured weekly) of the birds in Kg of $BW^{0.67}$, thus determining the maximum consumption. The variables evaluated were: daily feed intake (FI, g/bird), daily Lys intake (LysIntake, mg/bird), body weight (BW, kg), body weight change (BW, g/bird), daily egg production (EP, %/bird), egg weight, and daily deposition of Lys in egg mass (dLys, mg/bird), which was achieved by considering the concentration of 13% protein (*Ali, 2019*) and the 6.89% level of Lys in egg protein (*Ali, 2019*). Lys

**Table 1  Composition (g/kg) of the diets used in the lysine assay.**

| Ingredient (g/kg) | HPD[a] | NFD[b] |
|---|---|---|
| Corn | 356.97 | – |
| Soybean meal | 315.97 | – |
| Corn gluten meal (60% CP) | 181.22 | – |
| Soybean oil | 20.00 | 24.84 |
| Dicalcium phosphate | 10.13 | 15.02 |
| Limestone | 69.81 | 69.81 |
| Salt | 3.34 | 3.67 |
| Choline chloride (60%) | 0.84 | 3.40 |
| Mineral premix[c] | 0.25 | 0.25 |
| Vitamin premix[c] | 0.25 | 0.25 |
| DL-Met (99%) | 4.88 | – |
| L-Lys HCl (78%) | 5.72 | – |
| L-Thr | 2.71 | – |
| L-Val | 3.21 | – |
| L-Ile | 2.00 | – |
| L-Arg | 10.72 | – |
| LTrp | 1.81 | – |
| Potassium chloride | – | 11.95 |
| Corn starch | – | 249.03 |
| Sugar | – | 496.74 |
| Rice husks | – | 125.00 |

**Notes.**

[a] HPD, high protein diet.

[b] NFD, nitrogen free diet.

[c] Content per kg of the diet - vit A 6.668 IU; vit D3 1.668 IU; vit E 8 IU; vit K 3.2 mg; vit B1 1 mg; vit B2 3.34 mg; vit B6 2 mg; vit B12 5 mcg/kg; niacin 21 mg; chlorine 0.13 g; pantothenate acid 8 mg; folic acid 0.46 mg/kg; biotin 0.05 mg/kg; copper 8 mg/kg; iron 60 g; manganese 70 g; zinc 25 g; iodine 6.25 mg; selenium 0.12 mg.

mobilization was calculated from the change in BW, considering the mobilized protein fraction and, consequently, the proportion of Lys in the mobilized protein. Protein and Lys concentrations in the body were obtained from the method of a previous study (*Siqueira et al., 2021*).

## Description of responses by different mathematical functions

The variables dLys and LysIntake were related to the metabolic weight of the bird ($BW^{0.67}$). Two linear functions were used: linear regression and broken-line regression (Table 3).

To interpret the relationship between dLys and LysIntake, six e-functions were used, one of which was proposed in this research and the other five were obtained from the literature, considering the interpretation and biological meaning of the parameterization of the model (Table 3). The adjusted functions consisted of a monomolecular parameterized model with three (*Kebreab et al., 2008*; *Samadi Liebert, 2008*) and four parameters (*Kaps & Lamberson, 2004*; *Strathe et al., 2011*).

**Table 2  Nutritional levels of experimental diets.**

| Itens | HPD[a] | NFD[b] |
|---|---|---|
| Calculated composition (g/kg)[c] | | |
| Metabolizable energy (MJ/kg) | 12.5 | 12.5 |
| Calcium (g/kg) | 30.0 | 30.0 |
| Avaliable phosphorus (g/kg) | 3.0 | 3.0 |
| Analyzed composition (g/kg) | | |
| Crude protein | 350.0 | NI[e] |
| [d]Digestible Lys | 16.8 | NI |
| Digestible Met + Cys | 17.1 | NI |
| Digestible Met | 1.1 | NI |
| Digestible Trp | 0.3 | NI |
| Digestible Thr | 1.5 | NI |
| Digestible Arg | 2.5 | NI |
| Digestible Val | 1.7 | NI |
| Digestible Ile | 1.5 | NI |
| Digestible Phe | 1.9 | NI |

**Notes.**

[a] HPD, high protein diet.

[b] NFD, nitrogen free diet.

[c] The nutrient content of the ingredients used in the formulation was analyzed using a near-infrared spectrometer (NIR).

[d] The total amino acid content of the diets were analyzed HPLC and digestible content calculated using coefficients from *Rostagno et al. (2011)*.

[e] NI, Not identified.

## Model adjustment and selection statistics

The adjustment and selection statistics used were the determination coefficient ($R^2$), determination coefficient adjusted for the number of parameters ($R^2$ Adjust), Akaike information criterion (AIC), corrected Akaike information criterion (AICC) and the Bayesian information criterion (BIC), model quality was based on the lowest score for AIC, AICC and, BIC.

## Structure and assessment of linear and non-linear factorial models to estimate Lys intake based on BW and EO values

The factorial model calculated the nutrient Lys according to its partition, maintenance, and production. The nonlinear factorial model was based on the logarithmic transformation of *Samadi Liebert (2008)*, according to Eq. (9) (M9). In this model, the maintenance parameter was added after calculating the requirements for egg mass production.

$$\text{LysIntake} = BW^{0.67} \times [\text{Lysm} + (\ln R_{max} - \ln(R_{max} - 8.853 \times (EO/BW^{0.67})))/k] \quad (9)$$

The parameters necessary to calculate LysIntake were the $R_{max}$, Lysm, and k that were obtained from equation 3 (M3), equation 4 (M4), equation 5 (M5), equation 6 (M6), equation 7 (M7) and equation 8 (M8) in Table 3, generating the predicted values and the respective prediction errors for each monomolecular function.

To compare the LysIntake estimates by the nonlinear factorial model M9, the traditional factorial model (*Sakomura et al., 2015*; *Silva et al., 2019*) was used to estimate LysIntake

**Table 3  The functional forms used to describe the relationship between deposition of lysine (dLys) and lysine intake (LysIntake) daily.**

| Functional form | Function | Characteristic | Reference |
|---|---|---|---|
| $M1 = dLys = [\text{LysIntake-Lysm} \times BW^{0.67}]/a$ | Linear | Linear model, estimates the average requirement of the population. | *da Silva et al. (2019)* |
| $M2 = dLys = R_{max}+U \times (R\text{-LysIntake})$, for LysIntake$<$R | Linear | Broken line, estimates the average requirement of the population. | *Reis et al. (2018)* |
| $M3 = dLys = (R_{max} - R_{min})[1\text{-}e^{-k(\text{LysIntake}-\text{Lysm})}]$ | Exponential | Addition of the $R_{min}$ parameter with the response on the ordinate axis. | *Sousa et al. (2022)* |
| $M4 = dLys = R_{max}[1\text{-}e^{-k(\text{LysIntake}-\text{Lysm})}]$ | Exponential | The function does not provide the parameter of $R_{min}$. | *Kebreab et al. (2008)* |
| $M5 = dLys = R_{max}[1\text{-}e^{-k\text{LysIntake}}] - R_{min}$ | Exponential | The $R_{min}$ parameter with the response on the abscissa axis. | *Samadi Liebert (2008)* |
| $M6 = dLys = R_{min} + \text{Range}[1\text{-}e^{(-e-k(\text{LysIntake}-\text{Lysm}))}]$ | Exponential | It was a dual exponential model developed for the optimal response as a proportion of the asymptote. | *Strathe et al. (2011)* |
| $M7 = dLys = R_{max}[1\text{-}e^{(-e-k(\text{LysIntake}-\text{Lysm}))}]$ | Exponential | It is similar to model 6, with modified parameters. | *Strathe et al. (2011)* |
| $M8 = dLys = R_{max} - (R_{max} - R_{min})[e^{-k(\text{LysIntake}-\text{Lysm})}]$ | Exponential | This function was used to repair the Brody model. | *Kaps & Lamberson (2005)* |

**Notes.**

M, Model; Lysm, The daily intake of lysine for maintenance; BW, Body weight; a, The deposition of 1 mg Lys in the egg mass; Rmax, The maximum response for dLys (mg/kg $BW^{0.67}$); U, The rate of function growth; R, The estimated value of LysIntake for Rmax (mg/kg $BW^{0.67}$); Rmin, The minimum response for dLys (mg/kg $BW^{0.67}$); k, The rate of decay of the function.

according to Eq. (10) (M10).

$$\text{LysIntake} = BW^{0.67} \times [\text{Lysm} + a \times (8.853 \times EO)] \tag{10}$$

The parameters required to calculate LysIntake were Lysm and a, which were obtained from linear models M1 and M2.

The input variables in Eqs. (9) and (10) were BW and EO expressed in $kg^{0.67}$ andg/kg $BW^{0.67}$, respectively, and the value of 8.853 is the relationship between dLys and EO. LysIntake in Eqs. (9) and (10) is the model output of daily intake in mg/bird.

Assessment of dLys response prediction error as a function of LysIntake estimated by non-linear and linear factorial models. The prediction error was determined as the difference between the observed and predicted values of dLys. The errors were subjected to linear regression analysis according to the predicted value of a previous study (*St-Pierre, 2003*), according to Eq. (11) (M11).

$$ep = b0 + b1(Yp - \bar{Y}p) + \hat{e} \tag{11}$$

where $ep$ was the residual value for all observation; $b0$ and $b1$ were the estimates of the parameters, $Yp$ was the predicted value, $=Yp$ was the average of the predicted values, and $\hat{e}$ was the regression error of the residues to the predicted values. The decision rule was

**Table 4 Responses to lysine levels for daily feed intake, lysine intake, egg production, egg weight, egg mass, feed conversion ratio, lysine deposition, body weight, change body weight and lysine mobilization.**

| Lysine in diet | Feed intake | Egg production | Egg weight | Lysine intake | Egg mass | Feed conversion ratio | Lysine deposition in egg | Body weight | Change in body weight | Lysine mobilization |
|---|---|---|---|---|---|---|---|---|---|---|
| g/kg | g/bird | % | g | mg/bird | g/bird | g/g | mg/bird | kg | g/bird | mg/bird |
| 1.7 | 15.2 | 21.9 | 8.4 | 45.2 | 1.8 | 8.40 | 16.4 | 0.136 | −6.0 | −4.0 |
| 3.4 | 18.7 | 45.8 | 8.6 | 83.4 | 3.9 | 4.81 | 34.7 | 0.145 | −5.9 | −4.0 |
| 5.0 | 22.9 | 62.5 | 9.9 | 136.6 | 6.3 | 3.81 | 55.7 | 0.159 | −1.6 | −1.0 |
| 6.7 | 23.2 | 89.1 | 10.5 | 173.0 | 9.5 | 2.51 | 83.8 | 0.165 | 0.7 | 0.5 |
| 8.4 | 25.1 | 93.8 | 10.6 | 261.5 | 10.0 | 2.53 | 88.5 | 0.165 | −3.4 | −2.3 |
| 11.8 | 24.3 | 93.8 | 10.5 | 289.3 | 9.8 | 2.48 | 86.4 | 0.172 | −1.1 | −0.7 |
| 16.8 | 23.5 | 91.3 | 11.2 | 350.1 | 10.1 | 2.36 | 89.5 | 0.173 | 7.3 | 4.9 |
| General | 22.0 | 71.9 | 10.0 | 200.1 | 7.4 | 3.77 | 65.6 | 0.160 | −1.1 | −0.727 |
| SEM | 0.5 | 4.8 | 0.2 | 16.7 | 0.6 | 0.37 | 5.1 | 0.002 | 1.4 | 0.971 |
| *P*-Value | | | | | | | | | | |
| Treatment | <.0001 | <.0001 | <.0001 | <.0001 | <.0001 | <.0001 | <.0001 | <.0001 | 0.0683 | 0.1480 |
| Linear | <.0001 | <.0001 | <.0001 | <.0001 | <.0001 | <.0001 | <.0001 | <.0001 | 0.0043 | 0.0191 |
| Quadratic | <.0001 | 0.0002 | 0.0032 | 0.0051 | 0.0003 | <.0001 | <.0001 | 0.0283 | 0.5759 | 0.5877 |

**Notes.**
General, General average; SEM, The standard error of measurement.

based on the assumption that the model is impartial when the correlation approaches 1 and $R^2$ approaches 0. Therefore, the residues are not correlated with the predictions, and consequently, the value of b1 is close to zero for the unbiased model. The ratios of the parameters ($b0$ and $b1$) to regression error ($\hat{e}$), scalar error ($b0/\hat{e}$), and prediction bias ($b1/\hat{e}$) were obtained for the model.

## Statistical analyses

The assumptions of homoscedasticity and residual normality were tested. Subsequently, the data were subjected to analyses of variance, and when an invalid hypothesis was verified, the data were analyzed for linear and quadratic effects of the Lys levels, considering a significance of 0.05. The parameters of the models were estimated by the maximum probiosimilarity, using the NLMIXED procedure of SAS, considering the maximum random effect of the model (*Robbins, Saxton & Southern, 2006*). The values were calculated using SAS software (SAS Institute Inc., Cary, NC, USA, 2014, version 9.4).

## RESULTS

The Lys level in the diet affected the performance of Japanese quails breeders (Table 4), thereby rejecting the null hypothesis, where the variables do not differ with the levels of Lys in the diet ($P < 0.05$). The contrast analysis was significant for the linear and quadratic effects of Lys levels on bird replenishment, except for BW, which responded linearly to Lys levels in the diet. The homoscedasticity and residual normality were tested by the Shapiro–Wilk test, the data were normal, and the residuals are randomly distributed around zero ($p > 0.05$).

**Table 5  Fit statistics for the linear models, linear plateau and monomolecular functions for the relationship between deposition (Y) and lysine intake (X) of Japanese quail breeders.**

|    | Models | Regression | $R^2$ | $R^2$adj | AIC | AICC | BIC |
|----|--------|------------|-------|----------|-----|------|-----|
| M1 | Multiple linear | Linear | 0.710 | 0.690 | 215 | 216 | 212 |
| M2 | Linear plateau | Linear | 0.902 | 0.888 | 207 | 209 | 203 |
| M3 | Exponencial | Non linear | 0.840 | 0.810 | 219 | 221 | 213 |
| M4 | Exponencial | Non linear | 0.866 | 0.847 | 217 | 218 | 212 |
| M5 | Exponencial | Non linear | 0.866 | 0.846 | 217 | 218 | 212 |
| M6 | Exponential double | Non linear | 0.847 | 0.819 | 222 | 224 | 216 |
| M7 | Exponential double | Non linear | 0.841 | 0.818 | 207 | 208 | 203 |
| M8 | Exponencial | Non linear | 0.876 | 0.854 | 216 | 219 | 210 |

**Notes.**

$R^2$, R-Square; $R^2$adj, R-square adjust; AIC, Akaike Information Criterion; AICC, Corrected Akaike Information Criterion; BIC, Bayesian information criteria.

[M1] Model 1: $dLys = (LysIntake - 36 \times BW^{0.67})/3.69$.

[M2] Model 2: $dLys = 293 - 0.47 \times (682 - LysIntake)$.

[M3] Model 3: $dLys = (357 - 4) \times [1 - e^{(-0.0021 \times (LysIntake - 133))}]$.

[M4] Model 4: $dLys = 444 \times [1 - e^{(-0.0027 \times LysIntake)}] - 117$.

[M5] Model 5: $dLys = 327 \times [1 - e^{(-0.0027 \times (LysIntake - 114))}]$.

[M6] Model 6: $dLys = (119 + 170) \times [1 - e^{(-e^{(-1.025 \times (LysIntake - 511))})}]$.

[M7] Model 7: $dLys = 269 \times [1 - e^{(-e^{(-0.0041 \times (LysIntake - 374))})}]$.

[M8] Model 8: $dLys = 314 - (314 - 17) \times e^{[-0.0028 \times (LysIntake - 139)]}$.

The quails that were fed a lower level of Lys (1.7 g/kg) reduced their daily consumption by 39%, when compared with 8.4 g Lys per kg consumed the maximum value of 25.1 g/bird. The daily Lys intake at 1.7 g/kg was 13% of that of the highest level of Lys in the diet (16.8 g/kg). Therefore, the egg production and egg weight were reduced in different proportions. In 1.7 g/kg of Lys diet, the egg production decreased by 77% from the maximum value of 94%, while egg weight reduced only 25% of the maximum value of 11.2 g obtained at 16.8 g/kg of Lys diet (Table 4).

Egg mass and Lys deposition decreased by 82% in response to the limitation of Lys intake in the diet. Birds exhibited greater weight loss and consequently higher daily Lys mobilization values in diets with a greater degree of limitation in daily Lys intake. Increased intake of Lys linearly decreased its mobilization. Although consumption decreased, Lys limitation was responsible for low feed efficiency and consequently higher feed conversion values (Table 4). The feed conversion presented the largest amplitude (6.04) between the maximum (8.4 g/g at the level of 1.7 g/kg) and minimum (1.7 g/g at 16.8 g/kg) values corresponding to a change of 356% (Table 4). This result shows that the daily consumption of 15.2 g/bird would support a larger egg production, but Lys was limiting for protein synthesis.

## Analysis of adjustment and selection functions and statistics

The selection of Lys intake models is shown in Table 5. M2 presented better adjustment when considering only the values of the adjacent $R^2$. Model selection statistics (AIC, AICC, and BIC) indicated that the broken-line model (Table 5: M2) and the double exponential function (Table 5: M7) best adjusted the relationship between dLys and LysIntake.

The maintenance requirements of 36 and 139 mg/kg values of $BW^{0.67}$ obtained with M1 and M8, respectively, revealed no information on variability. A value of 52 mg/kg was

the $BW^{0.67}$ obtained for maintenance using M2 (Table 5) at dLys = 0; therefore, this also showed no variability. However, this model (Table 5: M2) presented a better adjustment and lower AIC, AICC, and BIC values (Table 5). In contrast, the M6 and M7 models (Table 5) estimated retention requirement values between 10 and 7 times greater than the M2-based value, respectively. The M7 (Table 5), along with the M2 presented better adjustments and lower values of AIC, AICC, and BIC (Table 5), while the $R_{max}$ estimates revealed the lowest determined value, which underestimated the genetic potential value, since the maximum response that was estimated as 269 mg/kg $BW^{0.67}$ was lower than the values obtained in the treatments with 6.7–16.8 g of Lys per kg.

For Models 3 and 4, the results of the adjustment and selection statistics support M4 as superior (Table 5). This model presented the highest maximum response value, which was estimated at 444 mg/kg $BW^{0.67}$. The maximum observed response of 297 mg/kg $BW^{0.67}$, was 67% of the estimated value for $R_{max}$. $R_{min}$ showed a variation of 43%, indicating a limited power of inference to interpret the animal response.

M3 presented estimates for $R_{max}$, $R_{min}$, k, and Lysm with smaller error values, supporting the biological significance in interpreting the bird response (Table 5). The maximum estimated response of 357 mg/kg $BW^{0.67}$ was 17% greater than the maximum observed value. The value of Lysm was 133 mg/kg $BW^{0.67}$, with a range of 128 to 137 mg/kg $BW^{0.67}$. Among the adjusted models with Lysm as a parameter, M5 returned 114 mg/kg $BW^{0.67}$ with a range of 25 to 203 mg/kg $BW^{0.67}$. For M8, 139 mg/kg was the estimated $BW^{0.67}$; therefore, the 133 mg/kg $BW^{0.67}$ value of the M3 was similar to those estimated in M5 and M8 (Table 5).

## Structure and assessment of linear and non-linear factorial models

The observed averages for LysIntake and dLys, in mg/kg $BW^{0.67}$ for each treatment and the respective estimated values are shown in Table 6. The estimates of the linear factorial models differed, especially in relation to the prediction of animal replenishment. M2 (Table 6) overestimated the response after ingestion of 286 mg/kg $BW^{0.67}$, while M1 presented better response estimates.

The M6 and M7 exponential models (Table 6) showed a discrepancy between the response estimates, where in M6, the estimated between levels did not differ and for M7 it was not possible to estimate for the level 6.7 g/kg of Lys per diet. While M3, M4, M5, and M8 revealed errors of 33.4, 33.4, 33.5, and 32.2 mg/kg $BW^{0.67}$, respectively (Fig. 1, Table 7). The prediction of nonlinear factorial models could only be reasonably evaluated with the aid of residue analysis (Figs. 1 and 2). Residue analysis statistics (Fig. 1) show that M4, M5 and M8 presented lower values for scalar error and prediction bias (Fig. 2). However, this analysis considers only the lines with observations, and some experimental units for M5 and M8 had dLys values greater than the $R_{max}$ of these models, resulting in negative values and therefore no solution, which decreased the number of observations for the analysis of the association between the residue and the predicted value, thereby limiting the use of these models in the factorial calculation of LysIntake.

M3 with the values of $R_{max}$, k and Lysm (LysIntake = 133+(ln(357)–ln(357–Deposition))/0.0021) and M4 (LysIntake = 117+(ln(444)-ln(444–Deposition))/0.0027)

**Table 6 Observed and estimated values of the lysine deposition by linear and nonlinear factorial models.**

| Observed and predicted variables | Lysine in diets, g/kg | | | | | | |
|---|---|---|---|---|---|---|---|
| | 1.7 | 3.4 | 5.0 | 6.7 | 8.4 | 11.8 | 16.8 |
| Observed variables | | | | | | | |
| Lysine intake | 190.1 | 333.8 | 484.0 | 586.4 | 889.3 | 955.3 | 1,155.7 |
| Lysine deposition in egg | 62.8 | 126.8 | 190.5 | 281.8 | 297.1 | 284.1 | 286.7 |
| Predicted variables | | | | | | | |
| Model 1: LysIntake $= 3.69 \times$ Deposition $+ 36 \times BW^{0.67}$ | | | | | | | |
| Predicted lysine intake | 241.0 | 477.7 | 713.5 | 1,050.7 | 1,106.9 | 1,059.3 | 1,069.2 |
| Predicted lysine deposition in egg | 53.8 | 96.4 | 140.9 | 171.3 | 261.5 | 281.0 | 340.6 |
| Error | 10.4 | 30.4 | 45.8 | 105.6 | 36.5 | 2.8 | $-49.9$ |
| Model 2: LysIntake $= 2.15 \times$ Deposition $+ 52 \times BW^{0.67}$ | | | | | | | |
| Predicted lysine intake | 148.5 | 286.8 | 424.8 | 621.4 | 654.2 | 626.6 | 632.7 |
| Predicted lysine deposition in egg | 82.1 | 148.6 | 218.1 | 265.5 | 406.4 | 436.9 | 530.1 |
| Error | $-17.2$ | $-21.8$ | $-33.4$ | 8.6 | $-107.9$ | $-153.3$ | $-237.1$ |
| Model 3: LysIntake $= 133+(ln\,(357)-ln\,(357-$Deposition$))/0.0021$ | | | | | | | |
| Predicted lysine intake | 225.4 | 342.4 | 503.4 | 879.2 | 1,043.3 | 911.7 | 945.3 |
| Predicted lysine deposition in egg | 39.8 | 120.4 | 183.9 | 216.4 | 280.8 | 289.9 | 311.4 |
| Error | 26.0 | 6.4 | 2.0 | 60.6 | 16.8 | $-5.8$ | $-23.5$ |
| Model 4: LysIntake $= 117+(ln\,(444)-ln\,(444-$Deposition$))/0.0027$ | | | | | | | |
| Predicted lysine intake | 173.6 | 241.7 | 327.0 | 490.7 | 533.7 | 498.6 | 507.2 |
| Predicted lysine deposition in egg | 61.1 | 145.4 | 206.5 | 235.5 | 286.7 | 293.0 | 307.1 |
| Error | 4.9 | $-18.6$ | $-20.3$ | 42.3 | 10.7 | $-8.9$ | $-19.7$ |
| Model 5: LysIntake $= 114+(ln\,(327)-ln\,(327-$Deposition$))/0.0027$ | | | | | | | |
| Predicted lysine intake | 193.3 | 296.1 | 446.0 | 857.2 | 994.1 | 924.3 | 997.3 |
| Predicted lysine deposition in egg | 60.6 | 145.0 | 206.3 | 235.3 | 286.6 | 292.9 | 307.1 |
| Error | 5.4 | $-18.3$ | $-20.1$ | 42.4 | 10.8 | $-8.9$ | $-19.6$ |
| Model 6: LysIntake $= 511+(ln\,(288)-ln\,(288-$Deposition$))/1.025$ | | | | | | | |
| Predicted lysine intake | 511.2 | 511.6 | 512.1 | 514.1 | 513.6 | 513.8 | 513.1 |
| Predicted lysine deposition in egg | 288.0 | 288.0 | 287.5 | 118.0 | 118.0 | 118.0 | 118.0 |
| Error | $-225.2$ | $-161.2$ | $-96.6$ | 163.8 | 179.1 | 166.1 | 168.7 |
| Model 7: LysIntake $= 374+(ln\,(269)-ln\,(269-$Deposition$))/0.0041$ | | | | | | | |
| Predicted lysine intake | 439.2 | 530.0 | 692.4 | – | 1,044.6 | 1,071.8 | 1,070.1 |
| Predicted lysine deposition in egg | 236.8 | 186.0 | 126.9 | 92.4 | 30.8 | 24.2 | 11.0 |
| Error | $-175.3$ | $-59.2$ | 68.4 | 194.5 | 265.9 | 259.9 | 275.1 |

Carvalho et al. (2023), *PeerJ*, DOI 10.7717/peerj.15637

**Table 6** (*continued*)

| Observed and predicted variables | Lysine in diets, g/kg | | | | | | |
|---|---|---|---|---|---|---|---|
| | **1.7** | **3.4** | **5.0** | **6.7** | **8.4** | **11.8** | **16.8** |
| Model 8: LysIntake = 139+(*ln* (314)-*ln* (314-Deposition))/0.0028 | | | | | | | |
| Predicted lysine intake | 219.0 | 324.2 | 482.3 | 974.3 | 1,382.5 | 1,161.4 | 1,096.2 |
| Predicted lysine deposition in egg | 56.4 | 140.5 | 200.7 | 228.8 | 277.6 | 283.5 | 296.5 |
| Error | 9.6 | −13.7 | −14.3 | 49.1 | 19.8 | 0.6 | −9.1 |

**Notes.**

Input variable: observed lysine intake, mg/kg of $BW^{0.67}$.

Output variable: deposition of lysine in the egg, mg/kg of $BW^{0.67}$.

Error: difference between observed and estimated for deposition of lysine in the egg.

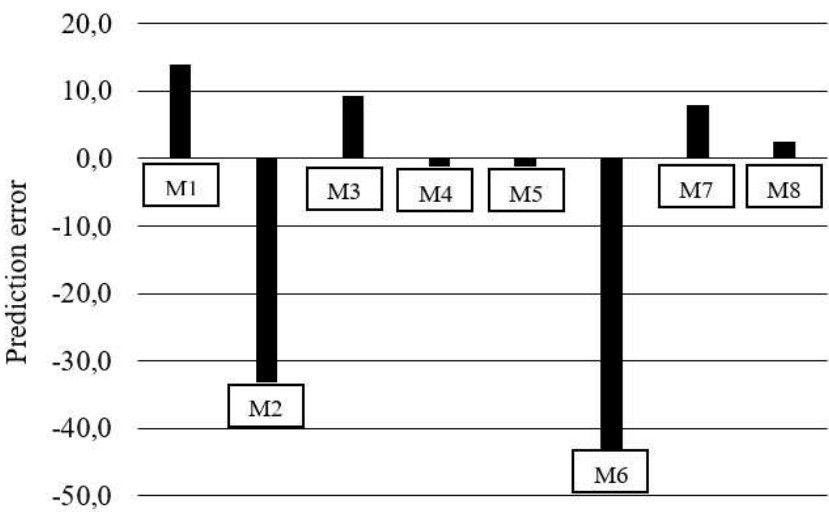

**Figure 1 Mean predicted of percentage lysine deposition errors for models (M).**

**Table 7 Statistics for assessment the error of prediction of the lysine deposition (Y - Ŷ) of Japanese quail breeders as a function of the lysine intake (X) calculated by the linear and non linear factorial model.**

|   | Models | Regression | e | b 0 | P-value | b 1 | P-value | R² | Scalar error | Prediction bias | 1-R² |
|---|--------|-----------|---|-----|---------|-----|---------|-----|--------------|-----------------|------|
| M1 | Multiple linear | Linear | 33.5 | 65.2 | 0.001 | −0.215 | 0.014 | 0.186 | 195 | 0.64 | 0.814 |
| M2 | Linear plateau | Linear | 28.7 | 66.2 | 0.001 | −0.497 | <.0001 | 0.749 | 231 | 1.73 | 0.251 |
| M3 | Exponencial | Non linear | 33.4 | 33.3 | 0.028 | −0.110 | 0.091 | 0.092 | 100 | 0.33 | 0.908 |
| M4 | Exponencial | Non linear | 33.4 | 0.2 | 0.990 | −0.011 | 0.869 | 0.001 | 1 | 0.03 | 0.999 |
| M5 | Exponencial | Non linear | 33.5 | 0.8 | 0.959 | −0.013 | 0.847 | 0.001 | 2 | 0.04 | 0.999 |
| M6 | Exponential double | Non linear | 35.2 | 403.1 | <.0001 | −1.978 | <.0001 | 0.953 | 1,146 | 5.62 | 0.047 |
| M7 | Exponential double | Non linear | 28.4 | 319.5 | <.0001 | −2.018 | <.0001 | 0.966 | 1,126 | 7.11 | 0.035 |
| M8 | Exponencial | Non linear | 32.2 | 2.2 | 0.890 | 0.014 | 0.842 | 0.001 | 7 | 0.04 | 0.999 |

presented no limitation when calculating LysIntake, but residue assessment statistics indicated a better predictive capacity for the factorial model with the M4 parameters. This result revealed that the biological interpretation and predictive capacity were not reconciled in the same model. M3 has parameters that assist the biological interpretation, but its application in the factorial model resulted in 10% less predictive capacity compared to M4 (Fig. 2, Table 7).

## DISCUSSION

To our knowledge, this is the first study to investigate the relationship between Lys levels and the response by Japanese quail breeders. The experimental period used here was 21 days, conforming with methodologies by *Silva et al. (2019)*. The experimental period could be reduced if there was greater differences in the diet levels (amplitude) of Lys. Both the present study and that of *Silva et al. (2019)* show that the definition of the treatments

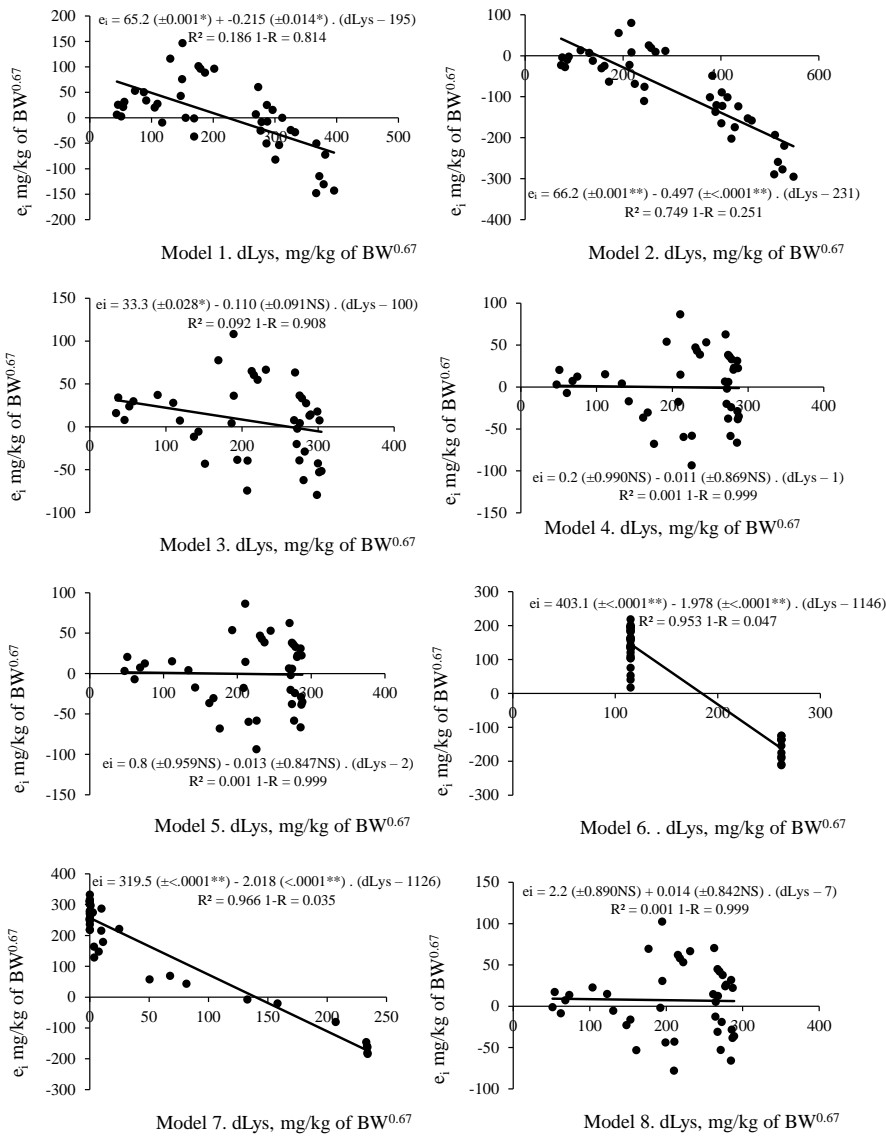

**Figure 2  Relationship between residual prediction (e_i) and predicted values for deposited lysine (dLys) by different models.** Model 1 and 2: linear and Model 4,5,6,7 and 8: non-linear. NS $p > 0.05$; ** $p < 0.01$.

and their amplitude should also be considered, along with the experimental period. The breadth of Lys levels and responses, especially in the egg production, is indispensable to support the findings independent of the statistical tool. Previous studies with Japanese quails obtained an amplitude of 4% (*Pinto et al., 2003*; *Costa et al., 2008*) and 8% (*Oliveira et al., 1999*), whereas this survey returned 77%. The results showed that the levels of Lys in the diet were limiting for Japanese quails breeders. The amplitude Lys deposition in the egg was close to 470%, supporting the findings of this study regarding the dietary limitation of Lys. In addition to experimental period reduced, was followed this study the "reduction",

formulated by Willian Russel and Rex Burch, which allowed the reduction in the number of animals used to maintain the precision of results (*Hubrecht & Carter, 2019*).

The feed of quails used in a breeding program should provide nutritional levels that support the expression of the maximum genetic potential. Therefore, linear procedures are limited to generating recommendations that approach the population average (*Baker, 1986*; *Basaglia et al., 2005*; *Hauschild, Pomar & Lovatto, 2010*; *Silva et al., 2019*) as opposed to the maximum. Among the variables analyzed here, the relationship between ingestion and deposition of Lys in the egg was selected to interpret the responses of the birds using exponential models. The results obtained in this study support the hypothesis that the method used can influence the Lys intake calculated and the interpretation of the animal response, especially the genetic potential of the bird, using six different exponential functions.

M2 was included in this analysis as a benchmark, especially for the interpretation of its parameters. The $R_{max}$ parameter of the broken-line model is associated with the average population potential (*Cosse & Baker, 1996*; *Hauschild, Pomar & Lovatto, 2010*) however, the $R_{max}$ values of the dual-exponential models (M6 and M7) were similar to that of the broken-line model. As both M6 and M7 models approximate the nutritional requirements of the average bird, the $R_{max}$ parameter population is forced to underperform and conform to the average bird population as their requirements are not being fully met. The parameterization of the exponential double was defined to (*Strathe et al., 2011*) approximate the asymptotic response of the model to the observed values, thereby avoiding the use of an asymptotic response ratio to establish optimal performance and necessary intake (*Strathe et al., 2011*). In dose–response studies, the use of proportions to establish optimal performance and the respective nutrient intake may vary from 50% to 95% of $R_{max}$ (*Halle, Jeroch & Gebhardt, 1984*; *Cosse & Baker, 1996*; *Samadi Liebert, 2008*; *Strathe et al., 2011*). Therefore, it represents a criterion that confuses the lack of model adjustment and the proportion of optimal performance. In an attempt to approximate the adjustment of the model to the data and the parameterization, the double exponential used could have limited the adjustment of the functions to studies with smaller amplitudes in the responses of nutrient deposition intake. In this study, the treatments vastly modified the responses of the birds by close to 476%, and the double exponential functions presented the poorest performance.

Two other objectives investigated in this research were the evaluation of the ability to interpret the response through the parameters of the model, and the predictive capacity when applied in a factorial approach. The results showed that it was not possible to reconcile the two objectives by the same mathematical model. The model that presented consistent estimation of the parameters and that aided in the interpretation of the response was the monomolecular with four parameters (M3). However, this model presented less predictive capacity when compared to the monomolecular with three parameters (M4). In a detailed analysis, the difference in the accuracy of these models (M3: 0.908 *vs.* M4: 0.999) is related to the scalar error, mainly the error of 26 mg/kg $BW^{0.67}$ in the M3 relative to the observed value at the first level of Lys in the diet, since the prediction bias value could only scarcely justify some differences between the models (Fig. 2).
The estimated $R_{min}$ parameter of 3.1 was close to zero (2–6 mg/kg $BW^{0.67}$), and this value has biological support. The lower level diet (1.7 g/kg) does not provide sufficient Lys for egg formation. Therefore, a significant body weight reduction was observed, equivalent to approximately 4 mg/kg $BW^{0.67}$ of Lys mobilized daily (Table 4). Subtracting the maintenance of 133 mg/kg $BW^{0.67}$, from the intake of 190 mg/kg $BW^{0.67}$ at the lowest level of Lys (1.7 g/kg), only 57 mg/kg $BW^{0.67}$ synthesis and deposition in the egg would be available. The prediction of the M4 of dLys and LysIntake at the lowest level was 61 and 174 mg/kg $BW^{0.67}$, respectively, resulting in 108% utilization efficiency, which indicates body reserve mobilization to sustain the minimal deposition of Lys in the egg. Based on this, the $R_{min}$ parameter estimated by M4 of 117 mg/kg $BW^{0.67}$ has no biological support, as it represents close to double the value seen in the diet with a lower level of Lys (63 mg/kg $BW^{0.67}$). Some authors attribute the interpretation of maintenance requirement to the $R_{min}$ of M4 considering that the value of $R_{min}$ represents an inevitable loss and must be provided in equal quantity by diet to avoid the animal undergoing a negative nitrogen balance. This finding reinforces the initial hypothesis that some factorial models use the value of the maintenance requirement extracted on the axis of the ordinate (*Strathe et al., 2011*; *Dorigam et al., 2014*; *Dorigam et al., 2017*), when the ideal is on the axis of the abscissa (*Kebreab et al., 2008*; *Silva et al., 2019*), to avoid confusion between minimal response, $R_{min}$, axis of the ordinate (*Silva et al., 2013*; *Dorigam et al., 2017*), and requirement of maintenance, Lysm, on the axis of the abscissa.

With the four-parameter monomolecular function, it was possible to estimate the maintenance requirement for Lys based on production responses close to zero. The use of curvilinear models for this purpose can be considered as a reasonable option, since for parameter estimation, all observations were used from the lowest to the highest level of Lys in the diet. When compared to the estimate of *Silva et al. (2019)* of 156.8 mg/kg $BW^{0.75}$, the figures appeared to differ, but in this research the metabolic weight was calculated using the $BW^{0.67}$, and *Silva et al. (2019)* used $BW^{0.75}$. When standardized the value of *Silva et al. (2019)* to the same basis used here: in the result is 136 mg/kg $BW^{0.67}$, considering a mean BW of 0.16 kg (Table 4), and this value is in the confidence interval of 128–137 mg/kg $BW^{0.67}$ estimated for Lysm in this survey.

The requirement for retention of quail breeders was 2.6 times greater than that of cut breeders (51 mg/kg $BW^{0.67}$) (*Silva et al., 2015b*) and 2.2 times greater than commercial dusts (61 mg/kg $BW^{0.67}$) (*Silva et al., 2015b*), demonstrating the difference between genotypes for egg production function, and thereby justifying this research.

Based on the factorial calculation of LysIntake and dLys (Table 6), it was possible to obtain the utilization efficiency of each level of Lys in the diet, with an average of 41% obtained with Model 3, and 87% for Model 4. The requirement of Lys per g egg mass calculated on the basis of these models was 23 mg/g for Model 3 and 11 mg/g for Model 4 considering the relationship between Lys deposition and use efficiency: 8.853/0.41 = 23 mg/g for Model 3 and 8.853/0.87 = 11 mg/g for Model 4. In previous studies, the efficiency of Lys was 47% (*Silva et al., 2019*), and Met + Cys, Thr, and Trp, returned values of 59%, 42%, and 26%, respectively (*Sarcinelli et al., 2020*). The mean of these results is 43%, which is similar to the average efficiency, considering all treatments, obtained with models M3

and M4, verifying the importance in the selection of the function to interpret and predict the animal response. Despite the similarity between the values found in this search (41%) and with the average (43%) obtained from previous studies (*Sarcinelli et al., 2020*; *Silva et al., 2019*), it is important to highlight the limitation of information on the concentration of amino acids contained in the quail egg, especially for tryptophan which was found in only one publication (*Ali, 2019*) and tritonin, whose concentration varied from 5.3 (*Ali, 2019*) to 7.3 mg/egg (*Genchev, 2012*). Therefore, establishing the amino acid profile of the quail egg will help to consolidate the understanding of the efficiency of amino acid use, since recent studies have reported that this efficiency by quails is half that of other layers (*Sakomura et al., 2015*; *Silva et al., 2015b*).

The daily Lys intake calculated by the non-linear factorial model was 284 mg/bird for a bird of 0.170 kg BW and daily production of 10 g/bird EO. To use the model, the first step is to change the values of BW ($0.305 = 0.170^{0.67}$) and EO ($32.8 = 10 \times 0.305$) to metabolic body weight (MBW). EO is then transformed to dLys (290 mg/kg $BW^{0.67}$), multiplying 32.8 by 8.853 (8.853 is the relationship between dLys and EO). To calculate LysIntake initially, only dLys (290 mg/kg $BW^{0.67}$) was used to obtain LysIntake in mg/kg $BW^{0.67}$: 931 mg/kg $BW^{0.67} = (133 + (ln(357) - ln(357 - 290))/0.0021)$, then multiplying by MBW (0.30) This model assumes solutions for dLys < 357 mg/kg $BW^{0.67}$, equivalent to 12.3 EO, which is the maximum egg mass production. Another limitation of this model relates to the diet, with a value of 0.0021 representing the rate of use of the dietary protein, based on the ingredients maize, soybean, and corn gluten, with 60%, necessitating the use of the proposed model with other ingredients.

The factorial model prediction was positioned based on the equation parameters in relation to the values found in the literature, which used studies with Japanese quail eggs, due to the absence of studies with breeders. The value of LysIntake for a bird with 0.170 kg BW and with daily production of 10 g/bird EO was 284 mg/bird. By the linear factorial model of *Rostagno et al. (2017)*, LysIntake was 267 mg/bird daily. The model of *Rostagno et al. (2017)* has been accepted by technicians and researchers in the area, and the difference shown here of 18 mg/bird may be a limiting factor for animals that are in genetic selection programs, especially considering the cumulative effect of the subdosage. Using the responses of 9.04 EO and 0.154 BW from the previous (*Pinto et al., 2003*) survey, LysIntake was calculated as 247 mg/bird using the non-linear factorial model proposed here, differing by 7 mg/bird from the value of 254 mg/bird (*Pinto et al., 2003*).

Therefore, the four-parameter monomolecular function proposed in this study is adequate for interpreting the animal response. The parameters of this function when used for non-linear factorial calculations were suitable for calculating lysine intake for Japanese quail breeders.

## CONCLUSIONS

The methodology used limited the supply of lysine and the birds responded to the degree of limitation, and the lysine response curve could be studied carefully. Considering the ability to interpret to predict the animal response the monomolecular function with four

parameters was balanced against the statistics of adjustment and selection of models, being a reasonable option. It was possible to estimate the requirement of lysine for maintenance $133 \pm 2$ mg/kg $BW^{0.67}$ andbased on average 41% efficiency the requirement of 22 mg Lys was obtained to produce 1 g egg output. The daily intake Lys calculated by the non-linear factorial model was 284 mg/bird for a bird with 0.170 kg BW and with daily production of 10 g/bird EO.

# ACKNOWLEDGEMENTS

We are thankful the Laboratory of Poultry Sciences of the Department of Animal Sciences and Veterinary, UNESP-Jaboticabal. We thank also the VICAMI by donation of quail's hens.

## Funding

This study was supported by the National Council for Scientific and Technological Development (CNPq) grant no. (432588/2016-7) and by the Coordenação de Aperfeiçoamento de Pessoal de Nível Superior Brasil (CAPES) financial support scholarship provided to the first author (code 001). The funders had no role in study design, data collection and analysis, decision to publish, or preparation of the manuscript.

## Grant Disclosures

The following grant information was disclosed by the authors:
National Council for Scientific and Technological Development (CNPq): 432588/2016-7.
Coordenação de Aperfeiçoamento de Pessoal de Nível Superior Brasil (CAPES): 001.

## Competing Interests

Tadia Emanuele Stivanin is employed by Vicami Codornas.

## Author Contributions

- Lizia Cordeiro de Carvalho conceived and designed the experiments, performed the experiments, prepared figures and/or tables, authored or reviewed drafts of the article, and approved the final draft.
- Manoela Sousa performed the experiments, authored or reviewed drafts of the article, and approved the final draft.
- Jaqueline Pavanini performed the experiments, authored or reviewed drafts of the article, and approved the final draft.
- Tadia Emanuele Stivanin performed the experiments, analyzed the data, authored or reviewed drafts of the article, and approved the final draft.
- Nelson José Peruzzi analyzed the data, authored or reviewed drafts of the article, and approved the final draft.
- Alan Rodrigo Panosso analyzed the data, authored or reviewed drafts of the article, and approved the final draft.

- Michele Lima conceived and designed the experiments, analyzed the data, prepared figures and/or tables, authored or reviewed drafts of the article, and approved the final draft.
- Edney Silva conceived and designed the experiments, analyzed the data, prepared figures and/or tables, authored or reviewed drafts of the article, and approved the final draft.

### Data Availability

The raw data is available in the Supplemental Files.

### Supplemental Information

Supplemental information for this article can be found online at http://dx.doi.org/10.7717/peerj.15637#supplemental-information.

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
