# Peer review of "Estimate of lysine nutritional requirements for Japanese quail breeders"

_PeerJ, doi:10.7717/peerj.15637_

## Round 0.1 · original submission · Major Revisions

Thank you for submitting your manuscript to PeerJ. Even though both reviewers have determined the revision to be minor, I will suggest it as "major" given the large number of comments that need to be addressed. Please make sure that you go through all the comments from both reviewers in addition to those listed below that are from me.

1. I highly recommend reaching out to professionals to improve the English proficiency of the manuscript as it could help make the ideas more easily understood and enhance the overall clarity of the work.

2. Table 2 was never mentioned throughout the entire text.

3. Regarding Table 3, 1) "General" and "SEM" in Table 3 need an annotation; 2) in Line 256, please explain what the null hypothesis is.

4. In Table 4, it should be "Non linear" not "Não Linear".

5. In Table 5, why does the independent variable LysIntake become the dependent variable in all eight models? Why do we need to predict LysIntake?

For Table 5, for the “error” row, please explain why some values seem to be missing and only have a dash "-" left. Also, the row name should be "error" not "erro". I suggest adding a figure to explain the errors in Table 5.

6. Please consider adding a scatter plot with the fitted value of all the models in Table 6.

Reviewer 1 ·

Basic reporting

1, This manuscript is still a draft version. A lot of terms are not consistent in different paragraphs and some numbers are not matched between the contents and the tables. For example, in line 30-31, the author mentioned monomolecular function but in line 459, the author used single-molecular function; in line 416, 22mg/g for Model 3 , then in line 417, 23 mg/g for model 3. In line 307, “after ingestion of 682 mg/kg BW…”, I cannot find the value in corresponding Table 5. In line 140, the author used “one mg Lys in the egg mas”, but later in line 149, the author used “1 mg of Lys in the egg mass”. So all those inconsistencies should be corrected.
2, There is no Key Words section.
3, In section 2.4 , description of responses by different mathematical functions and the discussion parts are very confusing.
In 2.4, for the functions (1) to (11), it is better to make a table and interpret each model in the context of this study and add the reference papers as one column. Also the model rationale should be demonstrated clearly, why all those functions are all needed and what are the differences, disadvantages and advantages of them.
In discussion, when the result number was used, interpret how you get the number, how you calculate and get the numbers according to the models in corresponding tables, and based on which model. For example, in line 434, “EO is then transformed to …, multiplying .. by …to calculate …” write down the model function here to make the steps clear.

Experimental design

I think this is good. However, as a reader, I will ask a general question, is that your sample a good representative of the corresponding bird population, Is that your final result very general result for a broader application ?

Validity of the findings

I tried to follow your findings, the main conclusion and the discussion are fine but there are a lot of words/descriptions are not clear and need to be updated and clearly interpreted. For example:
In line 309, “ a discrepancy between the response estimates, while … “ please specify what are the discrepancies.
In the 2.6 statistical analysis, the homoscedasticity and residual normality were tested, please provide the test result and p-values.
In line 257, “except for BW” is that also for other variables, like change in BW, Lysine Mobilization?
In line 259, “reduced their daily consumption by …” what is the specific daily consumption value? But later in line 273, “the daily consumption of 15.2 g/bird” was noted. Please make the readers easier to follow.
In line 189. “model 7 replaced Rmin + Range with Rmax”, if this is correct, then in function (6) it should be dLys = (Rmin + Range) [1- …].
In line 160 – 161, the notation for dLys and LysIntake are repeated.
In line 137, what does “a” mean in the function?

Additional comments

This study reports a novel finding to identify statistical relationship between the level of Lys and bird responses. This finding is interesting and potentially useful when considering the relationship between the supply of lysine and the bird’s response in real Japanese quail breeder applications. The authors used 8 models and discussed the different result and the advantage/disadvantages of the models.
In total, the rational of using 8 models and comparisons between those models, how much advantage of the optimal model is and why it can be used in more general bird breeder settings need to be further discussed. Also how much benefit it will make when using the final suggesting lysine intake level should also be investigated in the research.
Besides above, in session 2.4, the interpretation of the models should be improved to ensure the readers can clearly understand the application of the formular in the context of bird breeder.

Reviewer 2 ·

Basic reporting

The paper is relevant, that could bring novels insights about the nutritional requirements estimate. The propose of this study has a good basement, with a methodology well established.
Sometimes the discussion were confused, but I think that its possible to re-write.

The paper should be reviewed by an English language translator.

Experimental design

Maybe the experimental treatment be presented in a TABLE.

I all models it is necessary to describe de acronyms, for which one.

Describe how the criteria: AIC, BIC and AICC, could be interpreted?

Validity of the findings

It's important to clarify that a four parameters model is important for a berending program and how it is important ( estimates more accurate??).
The study was well designed and conducted. The results found were relevant and important for science and industry.

Additional comments

It is important to do a English language review. Some paragraphs were complicate to interpret because of the translation.

Annotated reviews are not available for download in order to protect the identity of reviewers who chose to remain anonymous.

---

## Round 0.2 · Minor Revisions

Thank you for addressing all the questions.

1. Please ensure that all titles are in the English language including Figure 2, Table 3, Table 4, Table 5, Table 7,

2. In the title of Table 6, the symbol of the error of prediction of the lysine deposition should be Y - Y_hat.

3. In Line 251, it says that "M5 and M8 presented lower values for scalar error and prediction bias (Figure 2)". What is the reason for not including M4 even though it exhibits lower prediction bias?

4. In Figure 2, Can you refresh my memory on which variable is plotted on the x-axis, and why different models are not utilizing identical x values?

Reviewer 1 ·

Basic reporting

based on previous comments:

1,
---- all singlemolecular have been updated to monomolecular;
---- all updated to 23 mg/g
Please check again to make sure the contents are clear and unambiguous.

2,
– added

3,
--- those equations are removed , please renumber the equations.

Experimental design

---- The authors discussed this in the discussion

Validity of the findings

most of the comments are addressed. and please make sure your contents and language is clearly interpreted.

---

## Round 0.3 · accepted · Accept

We appreciate your thorough attention to all the comments. We are pleased to inform you that we would like to accept the manuscript.

Reviewer 1 ·

Basic reporting

good structure and analysis results.

Experimental design

all content is improved after the second revise

Validity of the findings

great finding

Additional comments

Thanks for a good paper